# Rethinking Masked Autoencoders for Multi-Channel Fluorescence Microscopy: Adaptive Inter-Channel Masking

## Abstract

Pre-training on multi-channel images faces the challenge of exploiting complementary information across channels while preserving channel-specific features. Existing MAE-based masking strategies often struggle to balance these aspects, leading to suboptimal representations. We propose **Adaptive Inter-Channel Masking (AIM)**, a novel strategy for multi-channel MAE pre-training that combines *Channel-Specific Saliency Map (CSSM)* to enhance channel-specific representation learning and *Channel Complementary Masking (CCM)* to promote cross-channel information flow by preventing simultaneous masking at the same spatial location. To quantify these effects, we also introduce the *Channel Specificity Index (CSI)* and *Cross-Channel Interaction Index (CCI)*, capturing the degree of channel-specific encoding and cross-channel interaction. Our analysis reveals a trade-off between CSI and CCI in existing methods, whereas AIM achieves a balanced optimization, leading to consistent performance gains across multi-channel fluorescence imaging benchmarks. These results highlight the importance of balancing channel specificity and interaction for effective multi-channel representation learning and establish AIM as a principled masking strategy for multi-channel pre-training.

## 1 Introduction

Masked Autoencoders (MAE) (He et al., 2022; Huang et al., 2022; Peng et al., 2022; Shi et al., 2022; Wu & Mo, 2022; Cao et al., 2022) have emerged as a powerful self-supervised learning framework, achieving strong transfer performance on tasks such as image classification, semantic segmentation, and object detection by pre-training on large-scale unlabeled data. A key advantage of MAE is its ability to deliver competitive transfer performance with limited labeled samples, making it a practical and efficient strategy in diverse vision tasks. Motivated by this success, recent studies have extended MAE-style pre-training to domain-specific applications, including biomedical imaging (Kraus et al., 2024; Pham et al., 2025).

Unlike conventional RGB images, biomedical imaging data often consists of multiple independent channels, each capturing distinct biological signals. For instance, in fluorescence microscopy, different staining techniques are used to highlight specific organelles or protein expressions, and these channels collectively provide a comprehensive view of cellular structures. Multi-channel images thus share common spatial coordinates while encoding complementary biological information. This intrinsic property fundamentally differs from single-channel or RGB images, underscoring the need for representation learning strategies that simultaneously preserve channel-specificity and exploit cross-channel complementarity.

However, current pre-training approaches for multi-channel fluorescence microscopy images (Pham et al., 2025; Kraus et al., 2024) often adopt the same random masking strategy as standard MAE (He et al., 2022). Specifically, multiple channels are concatenated into a single input and subjected to random masking. This design overlooks the independence of channel-specific signals and frequently masks the same spatial locations across channels, thereby limiting cross-channel information exchange. To address this, channel-wise masking methods (Pham et al., 2025) have been proposed, where all tokens in channels are masked and reconstructed from the remaining channels. While this

encourages cross-channel learning, it also forces the model to depend exclusively on visible channels, making reconstruction more difficult and weakening the learning of channel-specific features. Moreover, both methods do not distinguish informative biological regions from background areas, further limiting their effectiveness in representation learning.

To address these challenges, we propose Adaptive Inter-Channel Masking (AIM), a novel masking strategy tailored for pre-training on multi-channel fluorescence images. AIM is designed to jointly enhance channel-specific representation learning and cross-channel interaction, enabling effective modeling of complex multi-channel structures. Specifically, AIM introduces two complementary techniques: (i) Channel-Specific Saliency Map (CSSM), which computes saliency scores independently for each channel based on token saliency and inter-channel differences, encouraging targeted channel-specific learning; and (ii) Channel Complementary Masking (CCM), which guarantees that at least one channel remains visible at every spatial position, thereby promoting cross-channel interactions by encouraging the use of information from other channels during reconstruction.

Furthermore, we introduce two novel metrics to quantify the essential aspects of multi-channel representation learning: Channel Specificity Index (CSI) and Cross-Channel Interaction Index (CCI). CSI measures the degree of orthogonality between channel embeddings. Lower cosine similarity corresponds to better separation of channel representations, resulting in higher CSI values that indicate effective channel-specific learning. CCI quantifies cross-channel information exchange by evaluating the attention weights assigned to masked tokens from visible tokens in other channels. Experimental results show that AIM simultaneously improves both CSI and CCI, and that downstream task performance is maximized when the two metrics are jointly improved. These findings suggest that CSI and CCI are not independent but complementary objectives, underscoring AIM as an effective pre-training strategy for multi-channel representation learning.

In summary, the main contributions of this study are as follows.

- **Adaptive Inter-Channel Masking (AIM):** A novel masking strategy for multi-channel fluorescence images that jointly improves *channel-specific representation learning* and *cross-channel interaction*.

- **Channel-Specific Saliency Map (CSSM):** A channel-specific saliency computation that identifies and uniquely salient regions in each channel, reinforcing channel-specific feature learning.

- **Channel Complementary Masking (CCM):** A complementary masking scheme that ensures at least one channel remains visible at each spatial position, thereby promoting effective cross-channel interaction during reconstruction.

- **New Evaluation Metrics:** Two quantitative measures, the *Channel Specificity Index (CSI)* and the *Cross-Channel Interaction Index (CCI)*, designed to assess the independence of channel representations and the effectiveness of cross-channel interaction.

## 2 RELATED WORK

**Multi-Channel Adaptive Models.** Various methods (Chen et al., 2023; Plummer et al., 2020; Savarese & Maire, 2019) have been employed for multi-channel image processing, with some approaches further extending to generate channel-dependent weights via hypernet (Ha et al., 2016). More recently, methods specifically designed for multi-channel image modeling have been proposed to adaptively leverage channel information. For example, ChAda-ViT (Bourriez et al., 2024) and ChannelViT (Bao et al., 2023) incorporate channel information with positional embeddings or introduce hierarchical channel sampling strategies. In particular, DiChaViT (Pham & Plummer, 2024) proposes a diverse channel sampling strategy combined with regularization techniques to enhance generalization across diverse channel configurations.

**Self-supervised Pretraining for Natural Images.** Following the success of Masked Image Modeling (MIM) (Huang et al., 2022; Dong et al., 2022; Xie et al., 2022) approaches such as MAE (He et al., 2022; Cao et al., 2022), a variety of works have explored how the choice of masking strategy (Kakogeorgiou et al., 2022; Wu & Mo, 2022; Liu et al., 2023) influences representation quality. Rather than relying solely on random masking, recent methods have introduced task- or semantics-aware schemes guide models toward semantically meaningful regions during pretrain-

ing. ADIOS (Shi et al., 2022) employs adversarial learning to mask regions with high reconstruction difficulty. SemMAE (Li et al., 2022) incorporates semantic priors to mask semantically rich patches. AMT (Liu et al., 2023) improves efficiency by pruning redundant patches with attention. Recently, SBAM (Choi et al., 2024) dynamically adjusts masking ratios using saliency information. These developments highlight the growing importance of adaptive and semantically guided masking strategies for effective representation learning.

**Pretraining for Multi-Channel and Microscopy Images.** In contrast to RGB images, multi-channel fluorescence microscopy (MCI) (Chen et al., 2023; Chandrasekaran et al., 2024) data consist of distinct biological channels, where each channel highlights specific organelles or proteins. Representation learning in this domain thus requires balancing channel-specific feature preservation with effective cross-channel interaction. Recent studies have extended MAE to this setting. CA-MAE (Kraus et al., 2024) adopts random patch masking with cross-attention decoders to integrate channel information, but it struggles to fully leverage spatial structures within each channel. Cha-MAEViT (Pham et al., 2025) introduces a shared decoder to alleviate this issue, yet still relies on random or channel-wise masking. DINO (Caron et al., 2021)-based approaches further propose channel-adaptive masking (Bourriez et al., 2024), where only a subset of channels is selectively masked during training, and channel embeddings are concatenated at the token level within the encoder, and this design better exploits spatial and channel-specific structures. Moreover, several studies have explored pretraining strategies tailored to pathology domains within the DINO framework, aiming to incorporate pathology-specific knowledge for improved representation learning (Kang et al., 2023; Chen et al., 2024).

## 3 PROBLEM STATEMENT AND PRELIMINARY

### 3.1 PRETRAINING ON MULTI-CHANNEL FLUORESCENCE IMAGES

Multi-channel images can be represented as $\mathbf{X} \in \mathbb{R}^{C \times H \times W}$, where $C$ denotes the number of channels and $H \times W$ represent the spatial resolution. Unlike RGB images, each channel $c$ in fluorescence microscopy captures a channel-specific biological signal $\mathbf{I}_c$ produced by distinct staining protocols, reflecting cellular structures, protein distributions, or gene expression patterns.

Since all channels share the same spatial coordinates while encoding different biological signals, masking-based pre-training must address two objectives simultaneously: reconstructing missing regions within each channel and exploiting complementary signals through channel interactions. Overemphasizing channel specificity risks ignoring inter-channel dependencies, whereas focusing only on cross-channel fusion may suppress biologically unique features. Thus, balancing channel-specific learning and cross-channel interaction forms the central challenge of multi-channel representation learning. Motivated by this, we propose an MAE-based masking strategy that explicitly preserves both channel uniqueness and inter-channel complementarity.

### 3.2 EXISTING MASKING STRATEGIES OF MULTI-CHANNEL IMAGES

**Random and Channel Masking on Multi-Channel Images.** We define two baseline masking strategies for multi-channel data with $C$ channels and $N$ tokens per channel ($N_C = N \cdot C$):

$$\mathcal{M}_{\text{rand}} = \{j \mid j \sim \mathcal{U}(1, N_C)\}, \ |\mathcal{M}_{\text{rand}}| = r \cdot N_C \qquad \mathcal{M}_{\text{ch}} = \{l \mid l \sim \mathcal{U}(1, C)\}, \ |\mathcal{M}_{\text{ch}}| = r \cdot C \quad (1)$$

Random masking (Kraus et al., 2024; Pham et al., 2025) samples individual tokens from the flattened sequence $\{x_1, \ldots, x_{N_C}\}$ at a fixed ratio $0 < r < 1$, producing a binary mask vector $\mathbf{M}_{\text{rand}} \in \{0, 1\}^{N_C}$, where $M_{rand,j} = 1$ indicates that token $x_j$ is masked. The masked input is obtained as;

$$X_{mask} = (1 - \mathbf{M}_{\text{rand}}) \odot X + \mathbf{M}_{\text{rand}} \odot \theta, \qquad (2)$$

where $\odot$ is an element-wise multiplication, and $X$ and $\theta$ are an input image and a learnable mask token, respectively. While simple and effective for natural images, this strategy ignores channel-wise independence and often masks the same spatial locations across channels, *limiting cross-channel interaction learning*. Channel masking (Pham et al., 2025), on the other hand, randomly selects channels at ratio $r$ and removes all tokens within those channels. This encourages reconstruction of missing channels from visible ones, promoting inter-channel information exchange. However, removing all tokens in the selected channels forces the model to over-rely on the remaining channels, *increasing reconstruction difficulty and weakening channel-specific feature learning*.

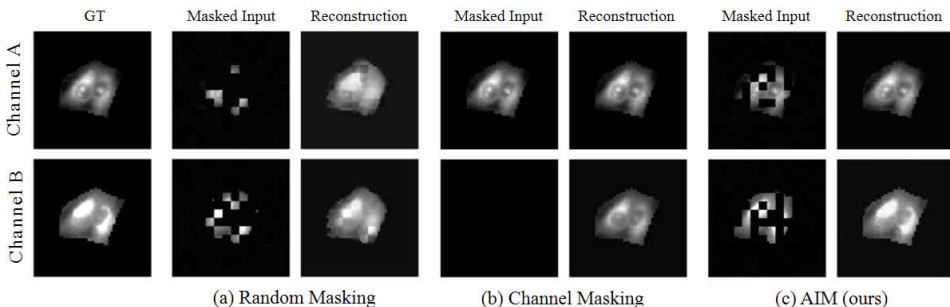

Figure 1: Comparison of masking strategies and reconstruction results: (a) **Random Masking**: Randomly masks patches in each channel; overlapping masks across channels often leave no visible cues, forcing the model to reconstruct using only intra-channel context. (b) **Channel Masking**: Entire patches in selected channels are masked, so the model relies solely on other channels and loses channel-specific information. (c) **AIM (ours)** applies **Channel Complementary Masking (CCM)** to ensure at least one channel remains visible at each spatial position, allowing the model to exploit complementary cross-channel information while preserving channel-specific details.

## 4 PROPOSED METHOD

### 4.1 MOTIVATION AND OVERVIEW

Effective representation learning in multi-channel fluorescence microscopy requires balancing two objectives: preserving channel-specific information while exploiting complementary cues through cross-channel interactions. However, existing masking strategies fail to achieve this balance, as discussed in Section 3.2. To address these limitations, we propose **Adaptive Inter-Channel Masking (AIM)**, a novel masking strategy that integrates Channel-Specific Saliency Map (CSSM) and Channel Complementary Masking (CCM). AIM first samples a subset of input channels for pre-training and then applies masking within those channels. CSSM computes channel-specific saliency to prioritize informative regions and reinforce channel-specific representation learning. CCM guarantees that at least one channel remains visible at each spatial position, preventing redundant masking and leveraging visible channels as reconstruction cues to promote cross-channel interaction. Through this design, AIM simultaneously enhances both channel-specificity and cross-channel interaction without introducing additional loss terms.

### 4.2 CHANNEL SAMPLING FOR MULTI-CHANNEL PRETRAINING

In multi-channel image environments, using all available channels simultaneously during pre-training can weaken channel-specific information. When tokens from all channels are mixed in a shared embedding space, strong cross-channel correlations dominate, overshadowing the distinct characteristics of individual channels. Sampling only a subset of channels at each iteration forces the model to represent the selected channels more distinctly, thereby preserving channel-specific information. Meanwhile, varying the subsets across training iterations exposes the model to diverse channel combinations, leading to more balanced cross-channel interaction. Prior studies have explored random selection of both subset size and composition (Bao et al., 2023), or fixed-size subsets throughout training (He et al., 2022; Kraus et al., 2024). Building on these insights, our analysis further confirms that pre-training with fewer channels can effectively preserve channel-specific information while enabling diverse cross-channel interactions. Consequently, we adopt a channel sampling strategy in which a subset $\mathcal{C}$ of channels is randomly chosen at each iteration. In our experiments, pre-training with two randomly selected channels (i.e., $|\mathcal{C}| = 2$) yielded the most effective performance, as further validated in the Appendix.

### 4.3 ADAPTIVE INTER-CHANNEL MASKING (AIM)

Given a sampled subset of channels $\mathcal{C}$, our masking strategy follows two principles: (1) informative tokens should be selected according to their importance to preserve channel-specific information,

and (2) masking must be complementary across channels so that at each spatial position, at least one channel remains visible.

To realize these principles, we introduce two complementary components: **Channel-Specific Saliency Map (CSSM)** extends saliency-based masking (Liu et al., 2023; Choi et al., 2024), originally developed for RGB images, to multi-channel fluorescence microscopy images by computing independent saliency maps per channel and applying relative saliency boosting to emphasize unique channel-specific patterns. **Channel Complementary Masking (CCM)** enforces cross-channel complementarity by ensuring that at least one channel remains visible at each spatial position. This allows visible channels to serve as reconstruction cues, facilitating effective cross-channel interaction.

Formally, the token selection process for the subset of channels $\mathcal{C}$ is formulated as:

$$M_i^c = \arg\max_{\tilde{M}_i^c} \sum_{c \in \mathcal{C}} \sum_{i \in \Omega} S_i^c \tilde{M}_i^c \quad \text{subject to} \sum_{c \in \mathcal{C}} M_i^c < |\mathcal{C}| \quad \forall i, \tag{3}$$

where a mask index $M_i^c = 1$ indicates that token $i$ in channel $c$ is masked, and $M_i^c = 0$ otherwise. $S_i^c$ denotes its saliency score (Sec. 4.3.1). $\Omega$ is the set of tokens per channel, $|\Omega| = N$. This objective maximizes total saliency of selected tokens under channel-exclusivity constraints. Since solving it exactly is computationally expensive, we adopt a greedy two-step approximation (Section 4.3.2).

### 4.3.1 CHANNEL-SPECIFIC SALIENCY MAP (CSSM)

**Saliency Computation for Individual Channels.** Masking tokens with higher importance is known to improve representation learning (Liu et al., 2023; Choi et al., 2024), but these approaches were originally designed for RGB images where the same saliency scores are assigned to all channels, limiting their applicability to multi-channel data. To preserve channel-specific importance, we extend the prior work (Choi et al., 2024) by computing saliency independently for each channel.

Given a set of tokens $\mathbf{T} \in \mathbb{R}^{N \times D}$, where $N$ is the number of tokens and $D$ the token dimension, the affinity matrix for channel $c \in \mathcal{C}$ is defined as: $\mathbf{P}^c = \sigma(\mathbf{T}^c(\mathbf{T}^c)^T)$, where $\mathbf{P}_{i,j}^c$ denotes the attention probability that token $i$ attends to token $j$, and $\sigma(\cdot)$ is a row-wise softmax function. The saliency score $\tilde{S}_i^c$ for each token $i$ is then obtained by summing over columns of $\mathbf{P}^c$ and applying min–max normalization $\text{Norm}(\cdot)$ for each channel independently:

$$\tilde{S}_i^c = \text{Norm}\left(\sum_{j=1}^{N} \mathbf{P}_{i,j}^c\right). \tag{4}$$

**CSSM based on Relative Saliency.** Single-channel saliency $S^c$ captures only intra-channel importance. Yet, a token with low intra-channel saliency may still hold relatively unique information compared to the same location in other channels. In particular, it is crucial to emphasize channel-specific information that uniquely appears in a given channel compared to others. To highlight this, we compute relative saliency as:

$$\Delta_i^c = \tilde{S}_i^c - \max_{c' \in \mathcal{C} \setminus \{c\}} \tilde{S}_i^{c'}, \tag{5}$$

and apply a boosted score:

$$S_i^c = \tilde{S}_i^c + \beta \cdot \text{ReLU}(\Delta_i^c) + \epsilon_i^c, \tag{6}$$

where $\beta$ controls the strength of channel-specific enhancement. $\epsilon_i^c \sim \mathcal{N}(0, \sigma^2)$ is added to the saliency scores to introduce stochasticity. This prevents the mask from clustering in specific spatial regions and thus yields more diverse masking patterns (Choi et al., 2024).

### 4.3.2 GREEDY SELECTION FOR CHANNEL COMPLEMENTARY MASKING (CCM)

To encourage effective cross-channel interaction, we introduce Channel Complementary Masking (CCM). It enforces that at least one channel remains visible at each spatial location, preventing all channels from being simultaneously masked. To this end, we present a greedy two-step solution to minimize (3).

**Step 1: Masking Ratio Adjustment.** Given an initial masking ratio $0 < r^c < 1$ for each channel $c \in \mathcal{C}$, we adjust it to satisfy the complementary constraint:

$$
\tilde{r}^c = \begin{cases} (|\mathcal{C}| - 1) \dfrac{r^c}{\sum_{k \in \mathcal{C}} r^k}, & \text{if } \sum_{k \in \mathcal{C}} r^k > |\mathcal{C}| - 1, \\ r^c, & \text{otherwise.} \end{cases}
\tag{7}
$$

This guarantees that no more than $(|\mathcal{C}| - 1)N$ tokens are masked in total, so that at least one channel remains visible per position through the subsequent masking process.

**Step 2: Saliency-Guided Complementary Masking.** For each channel $c$, we first select the top $K^c = N \times \tilde{r}^c$ tokens based on saliency scores $S_i^c$ to obtain a preliminary mask $M^c \in \{0,1\}^N$. If a token position $i$ is masked across all channels, *i.e.* $\sum_c M_i^c = |\mathcal{C}|$, we unmask the channel with the lowest saliency score at $i$ to satisfy the complementary constraint. The freed mask budget is reassigned to the next most salient unmasked token in the same channel.

**Discussion.** Through this design, CCM ensures that at every spatial position at least one channel remains visible. As a result, masked tokens in one channel can be reconstructed using visible tokens from other channels at the same location, which strengthens cross-channel interaction. In combination with CSSM, CCM achieves a balance between channel-specificity and cross-channel complementarity, leading to more effective multi-channel representation learning. Algorithmic details are provided in Appendix.

### 4.4 Implementation Details

**Shared Patch Embedding.** We divides the input image of each channel into patches and applies the same patch projection across all channels to generate patch embeddings.

**Dynamic Masking Ratio.** Following (Choi et al., 2024), we dynamically adjust the masking ratio $r^c$ for each channel $c \in \mathcal{C}$ based on the boosted saliency $S^c$ defined in (6).

**Architecture.** Our architecture consists of the Transformer Encoder and Decoder. In the encoder, patch tokens from all channels are concatenated along the sequence length dimension to form a single unified sequence. To encode channel identity explicitly, each channel $c$ is associated with a learnable embedding vector $\mathbf{ch\_emb}^c \in \mathbb{R}^D$. This embedding is added to every token in the channel: $\mathbf{z}_i^c = \mathbf{x}_i^c + \mathbf{ch\_emb}^c$, where $\mathbf{x}_i^c$ denotes the $i$-th token of channel $c$. Channel embeddings are initialized orthogonally to encourage independence across channels.

A single shared decoder is used for the sampled subset of channels $\mathcal{C}$. Masked and visible tokens from all channels are processed jointly, with channel embeddings again added to the decoder inputs to retain channel identity. Unlike per-channel decoders, the shared decoder provides strong parameter efficiency, scales easily to different numbers of channels.

## 5 Metrics for Channel-Specificity and Cross-Channel Interaction

To quantitatively evaluate the proposed masking strategy, we introduce two complementary metrics. The Channel-Specificity Index (CSI) measures how independently each channel forms representations, while the Cross-Channel Interaction Index (CCI) quantifies the extent to which information from other channels is utilized during reconstruction. These two metrics provide a quantitative framework for analyzing the structural limitations of existing masking strategies and for assessing how our method alleviates the trade-off between channel specificity and cross-channel interaction.

### 5.1 Cross-Channel Interaction Index (CCI)

The Cross-Channel Interaction Index (CCI) quantifies the extent to which masked tokens in the decoder rely on visible tokens from other channels during reconstruction. Let $M^c$ denote the set of masked tokens in channel $c$, and $V^{c'}$ the set of visible tokens in channel $c' \neq c$. Defining

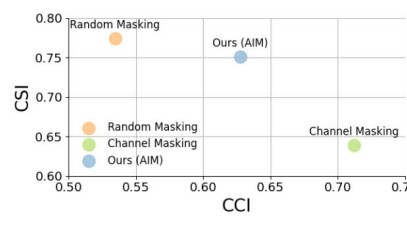 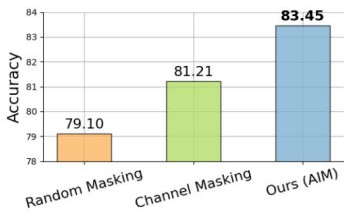

(a) CSI vs. CCI trade-off

(b) Downstream Classification Acc.

Figure 2: Comparison of three masking strategies: Random Masking, Channel Masking, and the proposed AIM. (Left) CSI vs. CCI trade-off plot shows that AIM achieves a balanced performance, improving CCI while maintaining competitive CSI compared to other methods. (Right) Downstream classification accuracy on JumpCP: AIM significantly outperforms both Random Masking and Channel Masking, achieving the highest accuracy of 83.45.

$V^{-c} = \bigcup_{c' \in \mathcal{C} \setminus \{c\}} V^{c'}$ where $\mathcal{C}$ indicates a sampled subset of channels, CCI is computed as:

$$\text{CCI} = \frac{1}{|\mathcal{C}|} \sum_{c \in \mathcal{C}} \frac{1}{|M^c|} \sum_{i \in M^c} \sum_{j \in V^{-c}} A_{ij}, \tag{8}$$

where $A_{ij}$ denotes the attention weight between a masked token $i$ in channel $c$ and a visible token $j$ from other channels in the decoder. A higher CCI indicates that masked tokens are reconstructed using information from other channels, reflecting more active cross-channel interaction.

## 5.2 CHANNEL SPECIFICITY INDEX (CSI)

The Channel Specificity Index (CSI) quantifies the independence of channel representations based on their channel embedding parameters. During MAE training, each channel $c$ is associated with a learnable channel embedding vector $\mathbf{e}_c \in \mathbb{R}^d$, which encodes channel identity. The degree of orthogonality between these vectors reflects how well the model preserves channel-specific information. Let $P = \{(c, c') \mid c \neq c', c \in \mathcal{C}, c' \in \mathcal{C}\}$ denote the set of all unordered channel pairs. For channel pair $(c, c')$, cosine similarity is $a_{c,c'} = \frac{\mathbf{e}_c \cdot \mathbf{e}_{c'}}{\|\mathbf{e}_c\| \cdot \|\mathbf{e}_{c'}\|}$, where $\mathbf{e}_c$ and $\mathbf{e}_{c'}$ are the channel embedding vectors. The CSI is then defined as:

$$\text{CSI} = 1 - \frac{1}{|P|} \sum_{(c,c') \in P} a_{c,c'}. \tag{9}$$

A higher CSI corresponds to lower average cosine similarity between channel embeddings, indicating better separation of channel representations and stronger channel-specificity.

## 5.3 MASKING STRATEGIES AND CCI–CSI ANALYSIS

Random masking (Kraus et al., 2024) and channel masking (Pham et al., 2025) exhibit an inherent trade-off between CCI and CSI, as illustrated in Fig. 2 (a). In random masking, each channel is only partially masked, so visible tokens within the same channel remain available. This self-referential property helps preserve channel-specific structures, leading to relatively higher CSI. However, because the same spatial positions can be masked across channels, cross-channel information exchange is suppressed, resulting in lower CCI. In channel masking, the model reconstructs completely masked channels using the remaining ones, which encourages cross-channel interaction and thus yields higher CCI. However, it prevents the model from directly learning their channel-specific properties, reducing CSI. In contrast, the proposed AIM integrates CSSM and CCM to enhance both CCI and CSI simultaneously. We further observe in Fig. 2 (b) that the best downstream performance is achieved when the two metrics are balanced.

## 6 EXPERIMENTS

### 6.1 EXPERIMENTAL SETUP

**Datasets.** We evaluated on two publicly available fluorescence microscopy benchmarks: JUMP-CP (Chandrasekaran et al., 2024), comprising 160 perturbations with images across eight channels

(five fluorescence, three brightfield) for perturbation detection, and CHAMMI (Chen et al., 2023), which integrates three multi-channel datasets (3–5 channels) with performance measured by the CHAMMI Score for domain generalization.

**Baselines.** We compare AIM against both *supervised* and *pretraining-based* models for multi-channel image analysis. **A. Supervised models:** AIM is compared with various *supervised* models for multi-channel image analysis, DepthwiseViT (Chen et al., 2023) processes each channel independently, TemplateMixingViT (Plummer et al., 2020; Savarese & Maire, 2019) learns channel-specific weights via shared templates, and HyperNetViT (Ha et al., 2016) generates channel-dependent weights using a hypernetwork. More recent models such as ChAda-ViT (Bourriez et al., 2024), ChannelViT (Bao et al., 2023), and DiChaViT (Pham & Plummer, 2024) further incorporate positional information, hierarchical or diverse channel sampling, and regularization to improve performance. **B. Pretraining-based models:** CA-MAE (Kraus et al., 2024) applies random masking to concatenated channels with channel-specific decoders, while Channel-MAE ViT (Pham et al., 2025) uses a shared decoder to reduce redundancy but still relies on random or channel-wise masking.

**Pretraining Setup.** Our approach employs the ViT-Small backbone with 21M parameters across all experiments. In the pretraining phase, the model is trained on the JUMP-CP (Chandrasekaran et al., 2024) dataset for 100 epochs using two decoder blocks and a batch size of 1024, keeping the default hyperparameter settings of MAE unless otherwise specified. The channel embeddings are initialized as orthogonal, consistent with DiChaViT (Pham & Plummer, 2024), while no additional Token Diversification Loss (TDL) is introduced.

**Downstream Tasks.** For downstream evaluation, we adhere to the training protocol of DiChaViT (Pham & Plummer, 2024). The AdamW optimizer is employed, with cross-entropy loss used for JUMP-CP and proxy loss for CHAMMI. The learning rate is linearly warmed up and then decayed following a cosine schedule. For all experiments, JUMP-CP is trained for 100 epochs, whereas CHAMMI is trained for 60 epochs under the same training settings.

**Evaluation Protocol.** To evaluate model performance, we measured the top-1 classification accuracy on the JUMP-CP. For the CHAMMI benchmark, We employed a 1-Nearest Neighbour classifier to compute the macro-average F1-score for each task individually, and then reported the average score on the WTC and HPA in Table 1, and present the detailed results in Table 5 of the Appendix.

**Experimental Setup for Reproduction.** All experiments employed the encoder architecture described in Section 4.4, pretrained on the JUMP-CP dataset and fine-tuned on the Diverse-Channel ViT benchmark. Since no official code is available at **Channel-MAE ViT** (Pham et al., 2025), we re-implemented the model using dynamic channel-patch masking, memory tokens, DCS, and Fourier loss to ensure a fair comparison focusing solely on pretraining masking strategies. For **Channel-Agnostic MAE** (Kraus et al., 2024), we follow the original configuration using a *separate decoder* and *Fourier loss* with a fixed *8-channel* training setup. Finally, **Channel Masking** (Pham et al., 2025), **Random Masking** (Kraus et al., 2024), and **AIM** are all reproduced under a fixed *2-channel* setup with a *shared decoder*, identical to the AIM methodology.

## 6.2 RESULTS

**Comparison with Supervised Methods.** Table 1 compares AIM with supervised baselines that are specifically designed for multi-channel image analysis and trained from scratch. These methods, such as DepthwiseViT, ChannelViT, and DiChaViT, incorporate architectural modifications to handle channel diversity but do not exploit large-scale pre-training. In contrast, AIM improves performance purely through its masking strategy while keeping the backbone unchanged. AIM achieves 83.45 top-1 accuracy on the JUMP-CP Full setting, exceeding the previous best (DiChaViT, 69.19) by 14.3 points. This highlights the effectiveness of our Adaptive Inter-Channel Masking strategy in capturing both channel-specificity and cross-channel interactions, enabling its transfer across tasks and leading to significant performance gains.

**Comparison with Multi-Channel Pre-Training Methods.** In Figure 3, we further compared AIM with recent multi-channel pre-training approaches, which typically rely on architectural modifications or channel-agnostic designs. Random masking partially induces cross-channel information exchange, but performance degradation occurs since the same spatial locations are often masked across multiple channels, reducing effective information utilization. Channel masking alleviates this issue

| Method | CHAMMI | JUMP-CP | |
| --- | --- | --- | --- |
| | Avg score | Full | Partial |
| HyperNetViT (Ha et al., 2016) | 56.08 | 47.07 | 42.43 |
| DepthwiseViT (Chen et al., 2023) | 61.80 | 49.86 | 44.98 |
| TemplateMixingViT (Plummer et al., 2020) | 58.16 | 52.48 | 43.85 |
| ChAda-ViT (Bourriez et al., 2024) | 63.93 | 65.03 | 42.15 |
| ChannelViT (Bao et al., 2023) | 64.97 | 67.51 | 56.49 |
| DiChaViT (Pham & Plummer, 2024) | 69.77 | 69.19 | 57.98 |
| **AIM (ours)** | **71.11** | **83.45** | **66.20** |

Table 1: Comparison of models on JUMP-CP and CHAMMI datasets. JUMP-CP uses top-1 accuracy. "Full" uses all channels; "Partial" uses a subset. All methods use Vision Transformer (Dosovitskiy et al., 2020) backbones.

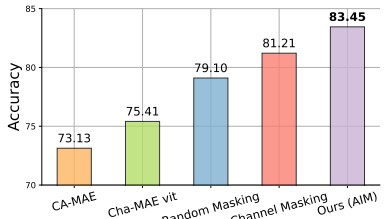

Figure 3: Performance on JUMP-CP Full. Baselines reproduced under our setup.

| Method | CSSM | CCM | Full |
| --- | --- | --- | --- |
| Random Masking | × | × | 79.10 |
| CSSM: $\beta = 0$ in (6) | o | × | 76.11 |
| CSSM only | o | × | 76.21 |
| CCM only | × | o | 82.97 |
| AIM: $\beta = 0$ in (6) | o | o | 83.29 |
| AIM (ours) | o | o | **83.45** |

Table 2: Ablation study on multi-channel microscopy classification (JUMP-CP) showing the effect of Channel-Specific Saliency Map (CSSM) and Channel Complementary Masking (CCM).

and yields performance gains, but still fails to adequately capture channel-specific information. Our approach avoids overlapping masks across channels through exclusive masking and complements it with channel-specific masking, thereby enhancing both channel specificity and cross-channel interaction. AIM improves representation quality purely through a masking strategy, without requiring additional components or losses. AIM consistently achieves the highest performance, demonstrating that the advantages of Adaptive Inter-Channel Masking extend beyond scratch-based learning and remain effective across diverse pre-training paradigms.

**Ablation Study.** Table 2 summarizes the ablation study on multi-channel microscopy classification (JUMP-CP). When only CSSM is applied, performance degrades compared to random masking. This result contrasts with prior findings in RGB image pretraining, *e.g.*, SBAM (Choi et al., 2024), where saliency-based masking improved performance by guiding models toward informative regions. In the multi-channel setting, salient patches from different channels often overlap at the same spatial locations, which suppresses cross-channel interaction even more severely than random masking when saliency-guided masking is applied. This observation highlights the importance of enforcing the complementary constraint, *i.e.* at least one channel remains visible at each spatial position. The benefit of this constraint is confirmed by the CCM-only result, which already improves performance to 82.97 by structurally preventing redundant masking across channels. When CSSM (with $\beta > 0$) is combined with CCM, the performance further increases to 83.45, demonstrating their complementary roles in enhancing both channel-specificity and cross-channel interaction. Even when $\beta = 0$ is used in AIM, performance (83.29) remains close to the full version but slightly worse, indicating that the relative saliency boost contributes positively even in the joint setting.

# 7 CONCLUSION

We presented **Adaptive Inter-Channel Masking (AIM)**, a pre-training framework for multi-channel fluorescence microscopy that integrates Channel-Specific Saliency Map (CSSM) and Channel Complementary Masking (CCM) to jointly enhance channel-specific representation learning and cross-channel interaction. To quantitatively assess these aspects, we introduced the Channel Specificity Index (CSI) and Cross-Channel Interaction Index (CCI). Experiments show that AIM achieves a favorable balance between the two, yielding consistent improvements across benchmarks. These findings emphasize the importance of explicitly modeling both channel specificity and inter-channel dependencies in the multi-channel pretraining, and suggest that AIM offers a generalizable foundation for structured multi-channel images beyond fluorescence microscopy.

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

# A APPENDIX

## A.1 ROBUSTNESS EVALUATION UNDER VARYING CHANNEL CONFIGURATIONS

As shown in Table 3 and Table 4, AIM (ours) consistently outperforms both ChannelViT and DiChaViT across different numbers of channels, demonstrating substantially higher robustness against channel dropping. Notably, dropping channel 3 causes the largest performance degradation for all methods; however, AIM (ours) experiences only a moderate decrease in accuracy (from 81.11% to 55.20%), indicating that it preserves informative representations more effectively and maintains stable performance even under severe channel reduction.

Table 3: Robustness evaluation under varying numbers of channels on the JUMP-CP classification task. Evaluation accuracy (%) across different numbers of channels. Mean $\pm$ standard deviation are reported.

| Method | 8 | 7 | 6 | 5 | 4 | 3 | 2 | 1 |
|---|---|---|---|---|---|---|---|---|
| ChannelViT (Bao et al., 2023) | 67.51 | 60.36±9.1 | 52.74±12.2 | 44.89±13.2 | 36.88±12.3 | 29.36±9.3 | 23.70±5.0 | 20.78±1.6 |
| DiChaViT (Pham & Plummer, 2024) | 69.19 | 61.91±9.3 | 54.49±12.4 | 46.35±13.4 | 38.00±12.4 | 30.09±9.3 | 23.97±4.9 | 20.90±1.6 |
| **Ours** | 83.45 | 76.52±8.9 | 68.51±12.3 | 59.07±14.3 | 48.28±14.4 | 36.89±12.1 | 26.96±7.1 | 21.55±2.3 |

Table 4: Robustness evaluation under sequential channel dropping on the JUMP-CP classification task. Evaluation accuracy (%) is reported as channels are progressively removed one at a time from the full 8-channel setting.

| Dropped Channel | ChannelViT | DiChaViT | AIM (ours) |
|---|---|---|---|
| 7 | 67.37 | 69.21 | 81.11 |
| 6 | 67.20 | 69.06 | 81.92 |
| 5 | 67.28 | 69.12 | 82.68 |
| 4 | 58.52 | 59.61 | 75.04 |
| 3 | 37.70 | 38.83 | 55.20 |
| 2 | 61.90 | 63.28 | 78.30 |
| 1 | 61.21 | 62.72 | 78.49 |
| 0 | 61.72 | 63.48 | 79.44 |

## A.2 PERFORMANCE COMPARISON ON THE CHAMMI DATASET ACROSS MULTIPLE TASKS

Table 5 summarizes the performance of various models on the CHAMMI (Chen et al., 2023) dataset across WTC (Viana et al., 2023), HPA (Thul et al., 2017), and CP (Way et al., 2022) tasks. We first pre-train the model using AIM (ours) and then fine-tune it on downstream tasks through the DiChaViT framework. Under this setting, AIM achieves the highest accuracy on most tasks, surpassing DiChaViT in all but three tasks. These results indicate that the cross-channel representations learned during AIM pre-training effectively transfer to downstream tasks when fine-tuned with the DiChaViT framework, highlighting its strength in multi-channel image analysis.

Table 5: Performance comparison of different models on the CHAMMI dataset across WTC, HPA, and CP tasks. Bold numbers indicate the best performance for each task. AIM (ours) consistently achieves superior results across most tasks, demonstrating its effectiveness in multi-channel image analysis.

| Model | WTC | | HPA | | | CP | | | |
|---|---|---|---|---|---|---|---|---|---|
| | Task1 | Task2 | Task1 | Task2 | Task3 | Task1 | Task2 | Task3 | Task4 |
| ChAda-ViT | 77.58 | 67.18 | 87.49 | 75.94 | 45.41 | 83.92 | 45.58 | **21.94** | 6.28 |
| ChannelViT | 78.36 | 67.58 | 83.93 | 76.73 | 47.97 | 77.70 | **55.16** | 21.89 | **6.38** |
| DiChaViT | 80.87 | **75.18** | 88.08 | 79.26 | 49.45 | 84.08 | 53.03 | 20.95 | 5.60 |
| **AIM (Ours)** | **83.49** | 74.31 | **90.18** | **82.62** | **51.20** | **84.98** | 52.46 | 19.80 | 6.15 |

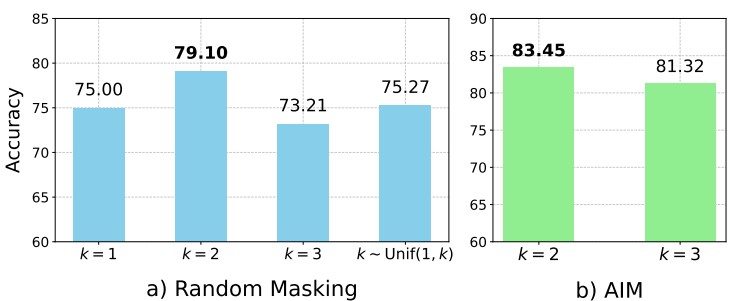

Figure 4: JUMP-CP Classification (Full) Results under Different Channel Selection Strategies

### A.3 COMPARISON WITH CHANNEL SAMPLING STRATEGIES

In multi-channel image environments where the number of channels can vary, channel sampling strategies remain a key challenge. To address this, we conduct a systematic experimental analysis on random masking and AIM (ours) to identify an effective strategy that balances reconstruction complexity and information diversity while ensuring computational efficiency. Our approach selectively utilizes a limited subset of two channels, enabling the model to capture the unique characteristics of each channel while effectively establishing cross-channel interaction.

We evaluate different channel sampling strategies under both **random masking** and **complementary masking** settings. In the random masking scenario, training with only one channel as the reconstruction target restricts cross-channel information flow, delaying the formation of dependencies and degrading performance. In contrast, using two channels jointly facilitates complementary information exchange, leading to better performance while maintaining channel-specific representation learning, as shown in Figure 4.

Under the **complementary masking** constraint, each masked token at a given spatial location can access reconstruction cues from at most $k-1$ visible channels. When $k = 2$, only one visible channel is available, resulting in the most challenging reconstruction objective. As $k$ increases ($k \geq 3$), masked tokens gain access to additional visible channels, simplifying reconstruction through richer cross-channel interactions. Our experiments reveal that the $k = 2$ setting, despite its difficulty, promotes stronger *channel-specific representation*, whereas larger $k$ values encourage *cross-channel interaction* at the expense of reduced channel specificity.

### A.4 ATTENTION MAP ANALYSIS UNDER DIFFERENT PRE-TRAINING STRATEGIES

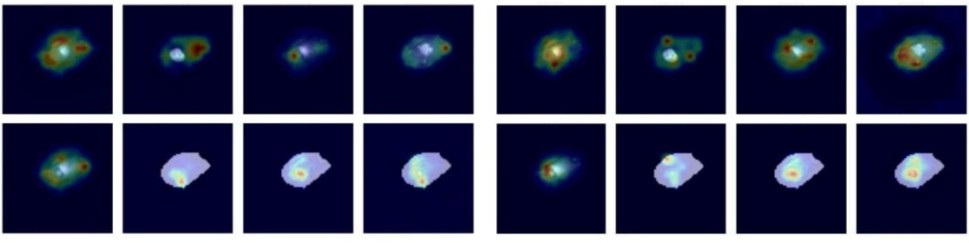

a) Random Masking       b) Adaptive Inter-Channel Masking (AIM)

Figure 5: Visualization of mean attention maps on the JUMP-CP classification task using models pre-trained with different masking strategies under the 2-channel setting.

We compare the mean attention maps of models pre-trained with Random Masking and AIM on the Jump-CP classification task (Figure 5). The Random Masking baseline shows overly concentrated attention on specific channels or spatial regions, leading to a noticeable reduction in attended areas. In contrast, AIM pre-training produces a more balanced and spatially diverse attention distribution across channels, indicating that the cross-channel information utilization learned by AIM transfers effectively to downstream tasks.

---

**Algorithm 1:** Greedy Selection for Channel Complementary Masking (CCM)

---

**Input:** Sampled channel set $\mathcal{C}$, token count per channel $N$,
per-channel saliency scores $\{S_i^{(c)}\}_{i=1}^N$ in (6) for all $c \in \mathcal{C}$,
initial masking ratios $\{r^{(c)}\}$ for all $c \in \mathcal{C}$.

**Output:** Binary masks $\{M_i^{(c)}\}_{i=1}^N$ for all $c \in \mathcal{C}$.

**Step 1: Masking Ratio Adjustment**       // ensure $\sum_{i,c} M_i^{(c)} \le (|\mathcal{C}| - 1)N$

**if** $\sum_{k \in \mathcal{C}} r^{(k)} > |\mathcal{C}| - 1$ **then**

$\quad \tilde{r}^{(c)} \leftarrow (|\mathcal{C}| - 1) \dfrac{r^{(c)}}{\sum_{k \in \mathcal{C}} r^{(k)}} \quad \forall c \in \mathcal{C}$

**else**

$\quad \tilde{r}^{(c)} \leftarrow r^{(c)} \quad \forall c \in \mathcal{C}$

**end**

$K^{(c)} \leftarrow \lfloor N \cdot \tilde{r}^{(c)} \rfloor \quad \forall c \in \mathcal{C}$

**Step 2: Saliency-Guided Complementary Masking**

**Step 2-1: Saliency-Guided Greedy Selection**       // preliminary masks

**for** $c \in \mathcal{C}$ **do**

$\quad \pi^{(c)} \leftarrow$ indices of tokens sorted by $S_i^{(c)}$ in descending order

$\quad \hat{M}_i^{(c)} \leftarrow 0$ for all $i$

$\quad$ **for** $j = 1$ **to** $K^{(c)}$ **do**

$\quad\quad \hat{M}_{\pi^{(c)}(j)}^{(c)} \leftarrow 1$

$\quad$ **end**

**end**

**Step 2-2: Complementary Refinement**       // no position masked by all channels

**for** $i = 1$ **to** $N$ **do**

$\quad$ **if** $\sum_{c \in \mathcal{C}} \hat{M}_i^{(c)} = |\mathcal{C}|$ **then**

$\quad\quad c^\dagger \leftarrow \arg\min_{c \in \mathcal{C}} S_i^{(c)}$       // least-salient at $i$

$\quad\quad \hat{M}_i^{(c^\dagger)} \leftarrow 0$       // unmask one channel at $i$

$\quad\quad$ // reassign freed budget in $c^\dagger$ to next most salient unmasked token

$\quad\quad$ Find smallest $j$ s.t. $\hat{M}_{\pi^{(c^\dagger)}(j)}^{(c^\dagger)} = 0$; set it to 1

$\quad$ **end**

**end**

**Return:** $M \leftarrow \hat{M}$

---

### A.5 ALGORITHM

Algorithm 1 illustrates the channel complementary masking. Given saliency scores across all channels, the algorithm first adjusts the masking ratios to ensure that at least one channel remains visible at every spatial position. For each channel, the top-$K^c$ tokens with the highest boosted saliency scores are selected for masking. If any spatial position is masked across all channels, the algorithm unmasks the channel with the lowest saliency score at that position and reassigns its masking budget to the next most salient unmasked token within the same channel. This complementary masking constraint guarantees that at every position, at least one channel remains visible, thereby enabling the reconstruction of masked tokens using information from the unmasked channels. Together with the masking ratio adjustment, this approach balances channel-specific representation learning with cross-channel interaction for more effective multi-channel pre-training.

## B  LLM USAGE DISCLOSURE

We used a large language model (ChatGPT, GPT-4) for retrieval and discovery purposes during the early stages of this work. Specifically, the model was used to:

- Suggest related works based on our research keywords and problem statement.
- Summarize abstracts and contributions of related papers to accelerate literature review.

The retrieved content was verified by the authors and not used verbatim in the paper. No LLM-generated content was used in the main writing or experimental sections. The authors take full responsibility for the final content of the paper.