# OpenReview forum: "Rethinking Masked Autoencoders for Multi-Channel Fluorescence Microscopy: Adaptive Inter-Channel Masking"
_ICLR.cc/2026/Conference — ICLR 2026 Conference Withdrawn Submission_

### Official Review · Reviewer_mWUQ · 2025-10-22

**Soundness:** 1
**Presentation:** 1
**Contribution:** 1
**Rating:** 0
**Confidence:** 5

**Summary:**

This paper claims an improved approach to MAE training in the context of multi-channel microscopy images by designing a new masking strategy oriented to the nature of such data. Their primary contribution is the channel complementary masking strategy (CCM) which offers the most significant improvements over random masking, and by combining CCM with channel-specific saliency map (CSSM) they arrive at a slightly better method which they call Adaptive Inter-channel Masking (AIM).

**Strengths:**

- The novel masking strategy appears to improve results over other masking strategies on the JUMP-CP dataset.
- The discussion of the tradeoff between their different metrics is interesting, and the ablation study is helpful for understanding their contribution.

**Weaknesses:**

The primary weakness is the overall lacking comprehensiveness of the evaluations and analysis in this work, which limits the ability to confidently establish their method as a novel contribution and a meaningful improvement over existing methods. I see the following weaknesses:

1. Compare the present work to the ChA-MAEViT paper [A] to observe the highly problematic nature of this submission in its current form. The present work has far less analysis with select results simply taken (and then partially masked) from [A]. Specifically, the only table of substantive results offered in the present work is in Table 1, and it seems that all of the values for the supervised models in this paper were simply copy-pasted directly from the results in Table 1 from [A] (minus the standard deviations, indicating a lossy copying). Except, there is one conspicuously missing result that was presented in [A], namely that paper's contribution -- i.e., the ChA-MAEViT model which obtains significantly better scores than AIM (it obtains 74.63 avg score on CHAMMI, 90.73 score on JUMP-CP Full, etc). Why is the core result from [A] neither included nor discussed? The authors seem to not have successfully reimplemented and reproduced the results from that paper, and then claim it is worse by showing a bar chart with it having less accuracy in Figure 3 (which differs from the reported accuracy in [A] by 15 points).

1. Furthermore, their method is evaluated on only two different datasets, but the comparisons across datasets appear incomplete. Specifically, it is unclear to me why Table 1 only includes AIM versus the supervised model, but then Figure 3 compares AIM to other MAEs. So in fact the only evidence provided that their new masking approach is a meaningful improvement is on one single metric (JUMP-CP accuracy) visualized in a bar chart in Figure 3 (a bar chart which is also strangely duplicated in Figure 2) - already this whole presentation is problematic for the reason indicated in point 1 above.

1. JUMP-CP is not a dataset: it is consortium of groups under the Broad institute that has produced some specific datasets. It is somewhat unclear what specific dataset is used here, and what the classification task actually is. The authors describe that there are 160 perturbations with various images, but the size and extent of the dataset is not clear here -- is the task perturbation classification? What kind of perturbations are these, siRNA? Other datasets would have been valuable to include in order to strengthen the evidence of their contribution here; for example, RxRx3-Core [B]. It also could have been fruitful to compare to the publicly available CA-MAE ViT-S, OpenPhenom (evaluated also in [B]), which was pretrained on both RxRx3 and on a selection of several million JUMP-CRISPR/ORF images [C].


A - Pham, Chau, Juan C. Caicedo, and Bryan A. Plummer. "ChA-MAEViT: Unifying Channel-Aware Masked Autoencoders and Multi-Channel Vision Transformers for Improved Cross-Channel Learning." arXiv preprint arXiv:2503.19331 (2025). NeurIPS 2025.

B - Kraus, Oren, et al. "RxRx3-core: benchmarking drug-target interactions in high-content microscopy." arXiv preprint arXiv:2503.20158 (2025). LMRL Workshop at ICLR 2025.

C - https://huggingface.co/recursionpharma/OpenPhenom

**Questions:**

For completeness, shouldn't all of the other MAE training strategies (random masking, CA-MAE, etc) be also included in Table 1? Indeed, it is interesting to see that even the apparently worst MAE on JUMP-CP (CA-MAE) is still better than the best supervised model in Table 1 on JUMP-CP (DiCHaViT). This is, however, in contrast to the results in [A] which suggest that a supervised loss was necessary to include for the CA-MAE. Why is this the case?

The authors do not say the patch size of their ViTs: does it employ 16x16 pixel patching?

**Details Of Ethics Concerns:**

The authors clearly copy-pasted the exact results in their Table 1 from the baseline supervised models (HyperNetViT up to DiChaViT, 434-437), results presented in another paper [A, Table 1], to the exact decimal, without indicating so. Furthermore, they exclude the best result from that work in order to present their contribution as an improvement (this paper's contribution actually underperforms compared to ChaMAEViT).

A - Pham, Chau, Juan C. Caicedo, and Bryan A. Plummer. "ChA-MAEViT: Unifying Channel-Aware Masked Autoencoders and Multi-Channel Vision Transformers for Improved Cross-Channel Learning." arXiv preprint arXiv:2503.19331 (2025). NeurIPS 2025.

---

### Official Review · Reviewer_3r8b · 2025-11-01

**Soundness:** 2
**Presentation:** 3
**Contribution:** 3
**Rating:** 6
**Confidence:** 4

**Summary:**

This paper proposes Adaptive Inter-Channel Masking (AIM), a new pre-training strategy for Masked Autoencoders (MAEs) on multi-channel microscopy images. The core problem with such images is that models must balance learning channel-specific features with exploiting cross-channel complementary information.

AIM addresses this via Channel-Specific Saliency Map (CSSM) and Channel Complementary Masking (CCM). The former identifies the most informative regions within each channel to reinforce learning unique, channel-specific features. The latter guarantees that at every spatial location, at least one channel remains visible. This forces the model to use information from other channels during reconstruction, promoting cross-channel interaction.

The authors also introduce two metrics to quantify this balance: the Channel Specificity Index (CSI) and the Cross-Channel Interaction Index (CCI). Experiments show that AIM successfully balances this trade-off.

**Strengths:**

This paper addresses a real issue in the field of image representation learning for microscopy images: modern image representation learning architectures are designed for natural RGB images, where there is high information overlap among all three channels, not images with significantly distinct information across channels. The authors are correct that an effective model must simultaneously preserve channel-specific features and exploit complementary information across different channels.

Here are more detailed strengths of the paper:

1. Using community-accepted benchmarks to evaluate results against other models
2. The CSSM and CCM are both intelligent answers to preserving channel-specific features and exploiting complementary information.
3. CSI and CCI are both well-designed metrics, useful for the community, that purport to measure a model's ability to do the above.

**Weaknesses:**

1. MAE's are a rather old architecture in visual representation learning that have not been state-of-the-art for many years in the open weights community; see DINOv2. You do not address this decision.

2. In Table 1, in the results section, the DINO-based methods are DiChaViT. Why not include DINOv2 itself for comparison?

Minor: You don't cite already existing work in this field, see [1].

[1] Kraus, Oren, et al. "Masked autoencoders are scalable learners of cellular morphology." arXiv preprint arXiv:2309.16064 (2023).

**Questions:**

1. Could this be extended to self-distillation strategies like DINOv2? DINOv2 has an iBOT component where AIM, of course, could be used, but I am more interested in the self-distillation (student-teacher) component.

2. You state that this work: "offers a generalizable foundation for structured multi-channel images beyond fluorescence microscopy". Yet, pthe aper provides no evidence or experiments on other types of multi-channel data (e.g., satellite imagery, medical MRIs) to support this broader claim. The ChannelVIT paper already has a satellite image benchmark dataset ready fro evaluation. Why did you decide not to include this despite many other details?

---

### Official Review · Reviewer_HeJH · 2025-11-02

**Soundness:** 3
**Presentation:** 2
**Contribution:** 2
**Rating:** 2
**Confidence:** 5

**Summary:**

The paper addresses the problem of multi-channel representation learning for microscopy imaging using masked autoencoders. The paper introduces a masking strategy that masks tokens from different channels depending on a saliency score and complementarity across channels. The paper introduces definitions and algorithms to improve cross channel interaction as well as preserve intra-channel information in the representations learned with ViTs and MAE. While the results indicate that performance improves empirically, the core contribution is incremental.

**Strengths:**

* The paper identifies an opportunity for improving masking strategies in MAE training: previous works employ random masking or channel masking, but not guided with saliency across channels.
* A saliency score is introduced and calculated for each channel individually, and then leveraged to identify tokens that can be more informative for learning.
* The formulations are presented with a detailed mathematical formulation.
* The analysis and experiments support the hypothesis and show improvements in performance compared to baseline methods.

**Weaknesses:**

* The method is trained with two channels at a time (line 210). The reader is pointed to the appendix for more information, but there is no quantitative evidence or analysis that suggests that two channels is sufficient.
* It is unclear how the masking ratios are initialized. Line 285: "CCM ensures that at every spatial position at least one channel reamins visible". With a high masking ratio r, is that even possible? For two channels, approximately 50% of channel would be masked. It is unclear how that changes with more channels.
* The proposed CCM algorithm has two steps, one for initializing the ratios and another one for selecting the tokens to be masked. According to the appendix, the ratios are not updated iteratively after the initialization presented in equation 7. Step two also seems sensitive to the order of sampling in the channels. The method is very heuristic, despite the effort to formalize everything with equations.
* CCI and CSI are metrics specific for MAE-type of models and do not provide information about how relevant the inter-channel interactions are for downstream analysis (task agnostic). These metrics are necessary analysis, but should not be considered a contribution because their potential for comparing other methods (e.g. fully supervised or distillation-based SSL) is limited.
* The baseline results in Table 1 come directly from the cited papers. Many ideas in this paper come from the paper (Pham et al. 2025), which was available in arXiv since March 2025. The Table should include results from that paper too. It is not scholarly to take inspiration and cite ideas from a paper and then ignore their results.
* Related to the above, the results from (Pham et al. 2025) are better than the results reported in Table 1. This invalidates the claim that the presented results are better than the previous best.
* The use of a single decoder and the idea of dynamic masking ratios also come from (Pham et al. 2025). Proper credit to these ideas should be given in context.
* Not sufficient analysis presented. The method introduces several parameters (e.g., sigma, beta, ratios). It is unclear how these parameters are set, and what the sensitivity of the model to these parameters is. No analysis presented on how these choices are made.
* Supervised masking was not considered. Prior work has used segmentation masks of cells to guide masking, indicating superior performance too (see https://www.biorxiv.org/content/10.1101/2024.12.06.627299v1). This is related to the proposed approach and was not evaluated.
* Other comments
Eq. 3 The indices i and c of the optimal solution M are reused in the argmax expression, which is is confusing. Do the authors mean to have different indices or perhaps no indices?
* Typos: "We divides [...] and applies" (line 294). "identifies and uniquely salient regions" (line 083)

**Questions:**

* Why were the results of Pham et al. (2025) not reported? How can the paper be framed in light of these results?
* Why there is no more analysis on training with more than 2 channels? Prior work has successfully considered multiple channels, and there is no strong evidence that this limitation is necessary.
* Is this specific to microscopy? Prior work has conducted experiments with other imaging types, and seem more general.

---

### Official Review · Reviewer_x98n · 2025-11-05

**Soundness:** 3
**Presentation:** 3
**Contribution:** 2
**Rating:** 4
**Confidence:** 4

**Summary:**

In this paper, the authors introduce a methodology for handling complementary information across channels while preserving channel-specificity in biological features when training single cell representation models using a Masked AutoEncoder objective.

To address existing challenges with random path masking and channel-wise masking, the authors propose an Adaptive Inter-Channel Masking (AIM) strategy that promotes both channel-specific salient information preservation while allowing efficient cross-channel interaction during pre-training. This is achieved by modifying the patch sampling procedure to sample salient regions of the cell (using a salient map while ignoring background or irrelevant information termed as CSSM) and Channel Complementary Masking where channel patches at the same spatial location are not masked at the same time.

To evaluate their contribution, the authors introduce a Channel Specificity Index and Cross-Channel Interaction Index to measure orthogonality between channel embeddings and how masked tokens use information from unmasked patches from other channels.

The authors conduct experiments on JUMP-CP and CHAMMI datasets and compare their method against several channel adaptive model baselines (both supervised and self-supervised). Using their methodology, they demonstrate improved accuracy in perturbation prediction task in JUMP-CP and CHAMMI over existing baselines.

**Strengths:**

* Provides a practical sampling protocol for handling single cell images that has a lot of uninformative patch tokens. Balances channel-specificity with cross-channel attention for multi-channel microscopy pretraining.
* Simpler drop-in sampling mechanism that's compatible with standard ViT-MAE setups. No additional loss functions or additional parameters.
* Introduces new metrics for measuring channel specificity and cross-channel interaction. Channel Specificity Index (CSI) and Cross-Channel Interaction (CCI) metrics
* Empirical results show large improvements in performance compared to baselines. Ablations showing CCM driving most of the gain while CSSM giving a smaller gain with combined sampling improving performance.

**Weaknesses:**

* CSSM depends on self-attention saliency. Early stage saliency maps can be noisy which might be a reason where CSSM-only might be hurting performance. The improvements in method (AIM) appears contingent on CCM.
* The methodology is highly specific to masked image modeling objectives and not generalizable to other pre-training objectives (or not demonstrated)
* All models are trained with a ViT-small backbone with a fixed patch size .Scalability of the methodology to larger models and fine-grained or coarser patch sizes is not explored.
* The experiments are limited to 2 datasets on classification task.
* The approach is fairly narrow in scope — it is demonstrated primarily on microscopy datasets, relies on a single pre-training framework (MAE), and is evaluated on a limited number of datasets using smaller model variants. While the empirical gains are promising, the contribution is more of a methodological tactic than a broadly generalizable framework.

**Questions:**

* Does the methodology hold consistently across different patch sizes - from fine grained to coarser patch tokens?
* How does the proposed method scale with an increased number of patch tokens and larger model sizes (i.e., more parameters)?

---

### Note · Authors · 2025-11-12

I have read and agree with the venue's withdrawal policy on behalf of myself and my co-authors.